# Factors associated with unskilled birth attendance among women in sub-Saharan Africa: A multivariate-geospatial analysis of demographic and health surveys

Isaac Yeboah Addo[1]*, Evelyn Acquah[2], Samuel H. Nyarko[3], Ebenezer N. K. Boateng [ID][4], Kwamena Sekyi Dickson [ID][5]*

**1** Centre for Social Research in Health, UNSW Sydney, Kensington, Australia, **2** Centre for Health Policy and Implementation Research, Institute of Health Research, University of Health and Allied Sciences, Volta Region, Ghana, **3** Department of Epidemiology, Human Genetics, and Environmental Sciences, University of Texas Health Science Center at Houston, Houston, TX, United States of America, **4** Department of Geography and Regional Planning, University of Cape Coast, Cape Coast, Ghana, **5** Department of Population and Health, University of Cape Coast, Cape Coast, Ghana

\* nadicx@gmail.com (KSD); i.addo@unsw.edu.au (IYA)

**Data Availability Statement:** AI relevant data are located at https://dhsprogram.com/data/available-datasets.cfm.

## Abstract

### Background

Several studies have shown that unskilled birth attendance is associated with maternal and neonatal morbidity, disability, and death in sub-Saharan Africa (SSA). However, little evidence exists on prevailing geospatial variations and the factors underscoring the patterns of unskilled birth attendance in the region. This study analysed the geospatial disparities and factors associated with unskilled birth attendance in SSA.

### Methods

The study is based on data from thirty (30) SSA countries captured in the latest (2010–2019) demographic and health surveys (DHS). A total of 200,736 women aged between 15–49 years were included in the study. Geospatial methods including spatial autocorrelation and hot spot analysis as well as logistic regression models were used to analyse the data.

### Results

There were random spatial variations in unskilled birth attendance in SSA, with the main hotspot located in Chad, whereas South Africa and the Democratic Republic of Congo showed coldspots. Residence (urban or rural), wealth status, education, maternal age at the time of the survey and age at birth, desire for birth, occupation, media exposure, distance to a health facility, antenatal care visits, and deaths of under-five children showed significant associations with unskilled birth attendance.

**Funding:** The author(s) received no specific funding for this work.

**Competing interests:** The authors have declared that no competing interests exist.

## Conclusion

Random geospatial disparities in unskilled birth attendance exist in SSA, coupled with various associated socio-demographic determinants. Specific geospatial hotspots of unskilled birth attendance in SSA can be targeted for specialised interventions to alleviate the prevailing disparities.

## Introduction

Historically, traditional or unskilled birth attendants defined as "persons who assist a mother during childbirth and who initially acquired their skills by delivering babies themselves or through apprenticeship to other traditional birth attendants" [1], have been the primary caregivers for women during pregnancy and childbirth in sub-Saharan Africa (SSA) [2]. In addition to their provision of care during pregnancy and child delivery, unskilled birth attendants play other roles, such as providing family planning information and advising pregnant women about their nutritional requirements [3]. Due to challenges associated with collecting maternal health data in some SSA countries [1], an aggregated estimate of unskilled birth attendance in the region is limited. However, current anecdotal reports show that more than one in two pregnant women (>50%) utilise the services of unskilled birth attendants in some SSA countries [4].

Employing the services of unskilled birth attendants in the management of pregnancy and child delivery is known to be associated with a high risk of maternal and neonatal morbidity, disability, and even death [5]. Unskilled birth attendants commonly lack the required knowledge, skills, or resources to balance risk with benefits during pregnancy and child delivery, and may have difficulty managing pregnancy or childbirth complications, such as haemorrhage, eclampsia, and obstructed labour [5]. It is therefore believed that women's utilisation of services from unskilled birth attendants during pregnancy and childbirth plays a significant contribution to the level of maternal and neonatal mortality in SSA which is ranked as the highest in the world with estimates of more than two-thirds of maternal deaths per annum or 68% or 533 maternal deaths per 100,000 live births, or 200,000 maternal deaths [6].

Given the assumption that many women in SSA continue to utilise the services of unskilled birth attendants rather than professional or skilled birth attendants, it is important to understand the factors accounting for these utilisations. Understanding these factors is essential for developing context-specific interventions tailored toward more specific population characteristics which in turn can inform efforts to reduce maternal deaths. However, to date, no empirical study, to the best of our knowledge, has examined the geospatial distribution and factors associated with the utilisation of maternal and neonatal services from unskilled birth attendants in the whole of SSA. Also, there are limited studies that have explored the spatial distribution of the utilisation of unskilled birth services in SSA. The available studies on unskilled birth attendants were mostly focused on only single countries in Africa, e.g., Olakunde, Adeyinka (5) for Nigeria, or explored perceptions of women about unskilled birth attendants from a qualitative perspective, e.g.,- Gurara, Muyldermans (7) for Ethiopia, and Selepe and Thomas (8) for South Africa [7–12].

Previous studies focused on skilled birth attendance among women in SSA have suggested that availability of a healthcare facility, distance between home and the healthcare facility, socio-economic status of pregnant women, place of residence, birth parity, ethnicity, previous contact with a healthcare system, and levels of education are associated with the use of services from skilled birth attendants during pregnancy and delivery [5, 13–16]. However, a significant

gap in current knowledge is the extent to which these factors apply to women who utilise the services of unskilled birth attendants. Exploring women's use of services from unskilled birth attendants and the factors associated with the use of such services will provide important supporting evidence for the development and implementation of evidence-based interventions. Using representative Demographic and Health Survey (DHS) data for women aged 15–49 years from thirty (30) SSA countries, this study examined factors associated with unskilled birth attendance among women in SSA. The study further examined the levels of service use from unskilled birth attendants among women in SSA as well as the hot and cold spot countries associated with the use of such services. Examining these factors can be useful to maternal health promoters, health practitioners, policymakers, and global health advocates addressing the unacceptable level of maternal mortality in SSA.

## Materials and methods

### Data source

The study analysed data from a pool of the latest demographic and health surveys (DHS) of thirty (30) countries in sub-Saharan Africa (SSA), conducted between 2010 and 2019. The surveys comprised nationally representative samples of women in their reproductive age groups (15–49 years), and the participants were selected based on a two-stage stratified cluster sampling procedure. The DHS generates reliable data on fertility, family planning, infant and child mortality, maternal and child health, among others, and was therefore deemed suitable for this study. Both standard and continuous DHS data were obtained for countries such as Angola, Burkina Faso, Burundi, Cameroon, Chad, Comoros, Congo, Cote D'Ivoire, Democratic Republic of Congo, Ethiopia, Gabon, Gambia, Ghana, Guinea, Kenya, Lesotho, Liberia, Malawi, Mali, Mozambique, Namibia, Nigeria, Rwanda, Sierra Leone, South Africa, Tanzania, Togo, Uganda, Zambia, and Zimbabwe. The unit of analysis comprised 200,736 women aged from 15 to 49 years selected from various households in these countries. The sample was limited to the last births of women within the five years preceding the surveys.

### Study variables and measurements

The outcome of interest was unskilled birth attendance. This was based on the person who assisted during the delivery of the last child. This comprised multiple categories of persons such as a doctor, nurse/midwife, community health officer/nurse, traditional birth attendant, traditional health volunteer, community/village health volunteer, relatives, and others. In this study, we created a binary outcome where deliveries assisted by traditional birth attendants, traditional health volunteers, community/village health volunteers, relatives, and others were measured as unskilled birth attendance while all others were considered skilled birth attendance [13, 17].

Additionally, various explanatory variables, of which most have been established by previous studies relating to the study outcome [17–19], were considered. These include age (Less than 20 years, 20–29 years, and 30–49 years), level of education (No education, Primary, Secondary, Higher), wealth status (Poorest, Poorer, Middle, Richer, Richest), marital status (Never in a union, Married, Cohabitation, Widowed, Divorced, Separated), occupation (Not working, Working), age at first birth (Less than 20 years, 20–29 years, 30–49 years), birth order (1, 2–3, 4 or more), mass media exposure (No, Yes), getting medical help for self: permission to go (Big problem, Not a big problem), getting medical help for self: getting the money needed for treatment (Big problem, Not a big problem), getting medical help for self: distance to health facility (Big problem, Not a big problem), desire for birth (Then, Later, No

more), antenatal care visits (Less than 4, 4 or more), under five mortality (No, Yes), place of residence (Urban, Rural) and country.

## Analytical procedure

We initially estimated the proportion of unskilled birth attendance by country and the socio-demographic characteristics of the respondents. We conducted a bivariate analysis of each explanatory variable and the outcome variable. All explanatory variables that had significant associations with the outcome variable were added to the multivariate model. Next, a multivariate analysis was performed using a logistic regression model to estimate associations between unskilled birth attendance and all the explanatory variables (age, level of education, wealth status, marital status, occupation, age at first birth, birth order, mass media exposure, getting medical help for self: permission to go, getting the money needed for treatment, distance to health facility, desire for birth, antenatal care visits, under-five mortality, place of residence and country variable). A multicollinearity test was performed for each variable which showed a mean-variance inflation factor (VIF) of 2.01 for the variables in the models. Unadjusted (Model 1) and adjusted odds ratios (Model 2) were then calculated for each variable at a 95% confidence interval. Stata (Version 14) was used to process and analyse the data. The results were sample-weighted to address any over-sampling and under-sampling in the total sample.

## Geospatial analysis

Regarding the geospatial analysis, the proportions data on unskilled birth attendance were merged with the shapefiles for each country using ArcMap version 10.5. Spatial autocorrelation (Global Moran's I) tool was used to explore the distribution of unskilled birth attendance in SSA after a hypothesis was set:

$H_1$: the distribution of unskilled birth attendance in SSA is not random

This spatial autocorrelation (Global Moran's I) tool was used to assess whether the distribution of unskilled birth attendance in SSA is random, clustered, or dispersed across the selected countries. The results to determine the type of distribution is dependent on the Moran's I index and the z-score or p-value. A positive and negative Moran's I index value implies a likelihood of clustering and dispersion respectively. The z-score or p-value indicates statistical significance, and the null hypothesis is accepted if the z-score is -1.65–1.65 and if the p-value is not statistically significant. In addition, a hotspot analysis (Getis-Ord G) was conducted to identify areas with relatively higher (hotspot) and lower (coldspot) occurrence of unskilled birth attendance among the selected countries based on z-score and p-value. A higher z-score and a lower p-value show a higher probability of a phenomenon occurring at a place. The Anselin Local Moran's I cluster, and outlier analysis were also conducted to ascertain statistically significant spatial outliers. This output showed outliers in the distribution using permutations such as high-high, low-low, high-low, low-high, and not significant.

## Results

### Unskilled birth attendance by country

Of the 200,736 women included in the study, the majority (8%) were from Burundi and the minority (1%) were from Lesotho. Chad had the highest proportion (69%) of women who utilised the services of unskilled birth attendants whereas the least proportion (3%) were from South Africa. Overall, about one in four women (27%) in SSA utilised the services of an unskilled birth attendant during pregnancy or delivery (Table 1).

**Table 1. Proportion of unskilled birth attendance in sub-Saharan Africa.**

| Country | Frequency | Percentage | Proportion of unskilled birth attendance |
|---|---|---|---|
| 1. Angola 2015–2016 | 8,577 | 4.3 | 50.0 |
| 2. Burkina Faso 2010 | 10,255 | 5.1 | 26.2 |
| 3. Burundi 2016–2017 | 16,316 | 8.1 | 7.9 |
| 4. Cameroon 2018 | 6,288 | 3.1 | 26.4 |
| 5. Chad 2014–2015 | 3,652 | 1.8 | 68.9 |
| 6. Comoros 2012 | 1,824 | 0.9 | 15.1 |
| 7. Congo 2011–2012 | 6,305 | 3.1 | 12.0 |
| 8. Congo DR 2013–2014 | 10,963 | 5.5 | 20.4 |
| 9. Cote d'Ivoire 2011–2012 | 5,304 | 2.6 | 40.8 |
| 10. Ethiopia 2016 | 7,052 | 3.5 | 60.4 |
| 11. Gabon 2012 | 3,986 | 2.0 | 15.9 |
| 12. Gambia 2013 | 5,677 | 2.8 | 16.8 |
| 13. Ghana 2014 | 4,201 | 2.1 | 25.9 |
| 14. Guinea 2018 | 5,362 | 2.7 | 44.2 |
| 15. Kenya 2014 | 6,896 | 3.4 | 40.1 |
| 16. Lesotho 2014–2015 | 995 | 0.5 | 18.5 |
| 17. Liberia 2013 | 4,186 | 2.1 | 16.0 |
| 18. Malawi 2015–2016 | 12,544 | 6.3 | 7.4 |
| 19. Mali 2018 | 6,157 | 3.1 | 29.5 |
| 20. Mozambique 2011 | 7,349 | 3.7 | 36.2 |
| 21. Namibia 2013 | 3,802 | 1.9 | 11.8 |
| 22. Nigeria 2018 | 15,091 | 7.5 | 46.1 |
| 23. Rwanda 2014–2015 | 5,796 | 2.9 | 7.5 |
| 24. Sierra Leone 2019 | 7,257 | 3.6 | 12.5 |
| 25. South Africa 2016 | 1,438 | 0.7 | 3.3 |
| 26. Tanzania 2015–2016 | 6,822 | 3.4 | 32.8 |
| 27. Togo 2013–2014 | 4,942 | 2.5 | 44.6 |
| 28. Uganda 2016 | 9,844 | 4.9 | 22.9 |
| 29. Zambia 2013–2014 | 7,188 | 3.6 | 18.3 |
| 30. Zimbabwe 2015 | 4,667 | 2.3 | 15.3 |
| All Countries | 200,736 | 100 | 26.7 |

## Unskilled birth attendance by socio-demographic characteristics of respondents

Over 40 percent of respondents without education, from the poorest households, and those who had less than four antenatal visits utilised the services of an unskilled birth attendant. More than one-third of respondents without media exposure (36.7%), with four or more children (34.3%), with permission (35.7%), and who were distanced from a health facility (36.5%) utilised services from an unskilled birth attendant. Additionally, about one-third of rural residents utilised the services of unskilled birth attendants (Table 2).

## Factors associated with unskilled birth attendance

The findings showed that place of residence, wealth status, age, level of education, marital status, occupation, age at first birth, birth order, mass media exposure, permission, getting the money needed for care, distance to a health facility, desire for birth, antenatal care visits,

**Table 2. Unskilled birth attendance by socio-demographic characteristics.**

| Variables | Frequency (n = 200,736) | Percentage | Proportion of unskilled birth attendance |
|---|---|---|---|
| *Place of residence* | | | |
| Urban | 63,953 | 31.9 | 12.1 |
| Rural | 136,783 | 68.1 | 33.6 |
| *Wealth status* | | | |
| Poorest | 48,435 | 24.1 | 43.7 |
| Poorer | 42,777 | 21.3 | 33.9 |
| Middle | 39,756 | 19.8 | 26.2 |
| Richer | 36,506 | 18.2 | 15.6 |
| Richest | 33,262 | 16.6 | 5.7 |
| *Age* | | | |
| Less than 20 | 15,121 | 7.5 | 25.2 |
| 20–29 | 96,268 | 48.0 | 25.1 |
| 30–49 | 89,347 | 44.5 | 28.8 |
| *Level of education* | | | |
| No education | 69,532 | 34.7 | 43.3 |
| Primary | 72,093 | 35.9 | 24.5 |
| Secondary | 52,465 | 26.1 | 10.9 |
| Higher | 6,646 | 3.3 | 3.0 |
| *Marital status* | | | |
| Never in union | 17,100 | 8.5 | 16.3 |
| Married | 131,305 | 65.4 | 29.1 |
| Cohabitation | 36,431 | 18.1 | 24.1 |
| Widowed | 3,141 | 1.6 | 31.9 |
| Divorced | 3,591 | 1.8 | 26.4 |
| Separated | 9,168 | 4.6 | 21.1 |
| *Occupation* | | | |
| Not working | 52,906 | 26.4 | 27.4 |
| Working | 147,830 | 73.6 | 26.5 |
| *Age at first birth* | | | |
| Less than 20 | 118,463 | 59.0 | 30.5 |
| 20–29 | 78,561 | 39.1 | 21.6 |
| 30–49 | 3,712 | 1.9 | 16.2 |
| *Birth order* | | | |
| 1 | 42,414 | 21.1 | 16.4 |
| 2–3 | 70,022 | 34.9 | 23.4 |
| 4 or more | 88,300 | 44.0 | 34.3 |
| *Mass media exposure* | | | |
| No | 78,388 | 39.1 | 36.7 |
| Yes | 122,348 | 60.9 | 20.4 |
| *Getting medical help for self: permission to go* | | | |
| Big problem | 37,391 | 18.6 | 35.7 |
| Not a big problem | 163,345 | 81.4 | 24.7 |
| *Getting medical help for self: getting money needed for treatment* | | | |
| Big problem | 111,763 | 55.7 | 31.7 |
| Not a big problem | 88,973 | 44.3 | 20.6 |
| *Getting medical help for self: distance to health facility* | | | |
| Big problem | 82,583 | 41.1 | 36.5 |

*(Continued)*

**Table 2.** (Continued)

| Variables | Frequency (n = 200,736) | Percentage | Proportion of unskilled birth attendance |
|---|---|---|---|
| Not a big problem | 118,153 | 58.9 | 19.9 |
| *Desire for birth* | | | |
| Then | 129,671 | 64.6 | 27.3 |
| Later | 12,237 | 6.1 | 31.8 |
| No more | 58,828 | 29.3 | 24.5 |
| *Antenatal care visits* | | | |
| Less than 4 | 85,250 | 42.5 | 40.5 |
| 4 or more | 115,486 | 57.5 | 16.6 |
| Under-five mortality | | | |
| No | 192,492 | 95.9 | 26.5 |
| Yes | 8,244 | 4.1 | 32.3 |

experience of death of an under-five child, and country of residence had a significant relationship with unskilled birth attendance (Table 3).

A higher likelihood of utilising the services of unskilled birth attendants was observed among rural women (AOR = 1.90, CI = 1.83, 1.97) compared to urban women. Those with the poorest wealth status were more likely (AOR = 4.36, CI = 4.09, 4.65) to utilise the services of unskilled birth attendants compared to those with the richest wealth status. Women who had desired to have children later (OR = 1.17, CI = 1.13, 1.20) and those who had a big problem getting money needed for care (OR = 1.05, CI = 0.1.02, 1.09) had a higher likelihood of utilising the services of unskilled birth attendants compared to those who desired to have children then and those who did not have a big problem getting money needed for care (Table 3).

Our findings also showed that teenagers (less than 20 years) had a higher likelihood (AOR = 1.11, CI = 1.05, 1.17) of utilising the services of unskilled birth attendants during delivery compared to women aged 20–29 years. Women who had their first birth at age 30–49 years (AOR = 0.81, CI = 0.72, 0.91) were less likely to utilise the services of unskilled birth attendants compared to those who had their first birth when they were less than 20 years of age. Women with no formal education had higher odds (AOR = 1.57, CI = 1.52, 1.62) of utilising the services of unskilled birth attendants during delivery compared with those with primary education. Women who were not exposed to mass media had a higher likelihood (AOR = 1.25, CI = 1.22, 1.29) of utilising the services of unskilled birth attendants during delivery compared to those who were exposed to mass media (Table 3).

Furthermore, the women who saw distance to a healthcare facility as a big problem had a higher likelihood (AOR = 1.49, CI = 1.45, 1.53) of utilising the services of unskilled birth attendants during delivery compared to women who stated that distance to healthcare facilities was not a big problem. Women who had less than 4 antenatal care visits had higher odds (AOR = 1.10, CI = 1.04, 1.17) of utilising the services of unskilled birth attendants during delivery compared to women who had 4 or more antenatal care visits. Women who had experienced the demise of their under-five child were more likely (AOR = 1.10, CI = 1.04, 1.17) to utilise the services of unskilled birth attendants during delivery compared to those who had not experienced the death of their under-five child (Table 3).

## Spatial distribution

Results from the spatial autocorrelation analysis (Fig 1) revealed that the spatial variations in unskilled birth attendance in sub-Saharan Africa (SSA) were random. Therefore, the alternate

**Table 3. Logistic regression of factors associated with unskilled birth attendance.**

| Variables | Model 1 | Model 2 |
|---|---|---|
| | **Odds Ratio** | **Adjusted Odds Ratio** |
| | **(95% Confidence interval)** | **(95% Confidence interval)** |
| *Place of residence* | | |
| Urban | Ref | Ref |
| Rural | 3.67***(3.58, 3.77) | 1.90***(1.83, 1.97) |
| *Wealth status* | | |
| Poorest | 12.76***(12.14, 13.40) | 4.36***(4.09, 4.65) |
| Poorer | 8.42***(8.01, 8.86) | 3.17***(2.98, 3.37) |
| Middle | 5.83***(5.54, 6.14) | 2.49***(2.34, 2.64) |
| Richer | 3.03****(2.87, 3.19) | 1.73***(1.62, 1.84) |
| Richest | Ref | Ref |
| *Age* | | |
| Less than 20 | 1.01(0.97, 1.05) | 1.11***(1.05, 1.17) |
| 20–29 | Ref | Ref |
| 30–49 | 1.21***(1.18, 1.23) | 0.92***(0.89, 0.95) |
| *Level of education* | | |
| No education | 2.36***(2.31, 2.41) | 1.57***(1.52, 1.62) |
| Primary | Ref | Ref |
| Secondary | 0.38***(0.37, 0.39) | 0.58***(0.56, 0.61) |
| Higher | 0.10***(0.08, 0.11) | 0.26***(0.22, 0.30) |
| *Marital status* | | |
| Never in union | 0.47***(0.45, 0.49) | 0.97(0.92, 1.03) |
| Married | Ref | Ref |
| Cohabitation | 0.77***(0.75, 0.79) | 0.95**(0.91, 0.98) |
| Widowed | 1.14**(1.06, 1.23) | 1.14**(1.04, 1.25) |
| Divorced | 0.87***(0.81, 0.94) | 1.11*(1.01, 1.22) |
| Separated | 0.65***(0.62, 0.68) | 0.95(0.89, 1.01) |
| *Occupation* | | |
| Not working | 1.04***(1.02 | 1.03*(1.00, 1.07) |
| Working | Ref | Ref |
| *Age at first birth* | | |
| Less than 20 | Ref | Ref |
| 20–29 | 0.63***(0.62, 0.64) | 0.90***(0.87, 0.92) |
| 30–49 | 0.44***(0.40, 0.48) | 0.81***(0.72, 0.91) |
| *Birth order* | | |
| 1 | 0.38***(0.36, 0.39) | 0.51***(0.49, 0.54) |
| 2–3 | 0.59***(0.57, 0.60) | 0.79***(0.76, 0.82) |
| 4 or more | Ref | Ref |
| *Mass media exposure* | | |
| No | 2.26***(2.21, 2.30) | 1.25***(1.22, 1.29) |
| Yes | Ref | Ref |
| *Getting medical help for self: permission to go* | | |
| Big problem | 1.69***(1.65, 1.73) | 1.02(0.98, 1.05) |
| Not a big problem | Ref | Ref |
| *Getting medical help for self: getting money needed for treatment* | | |
| Big problem | 1.79***(1.76, 1.83) | 1.05***(1.02, 1.09) |

(*Continued*)

**Table 3.** (Continued)

| Variables | Model 1 | Model 2 |
| --- | --- | --- |
| | Odds Ratio | Adjusted Odds Ratio |
| | (95% Confidence interval) | (95% Confidence interval) |
| Not a big problem | Ref | Ref |
| *Getting medical help for self: distance to health facility* | | |
| Big problem | 2.31***(2.26, 2.35) | 1.49***(1.45, 1.53) |
| Not a big problem | Ref | Ref |
| *Desire for birth* | | |
| Then | 1.16***(1.13, 1.18) | 1,17***(1.13, 1.20) |
| Later | 1.44***(1.38, 1.50) | 1.11***(1.06, 1.17) |
| No more | Ref | Ref |
| *Antenatal care visits* | | |
| Less than 4 | 3.41***(3.34, 3.48) | 2.61***(2.54, 2.68) |
| 4 or more | Ref | Ref |
| *Under five mortality* | | |
| No | Ref | Ref |
| Yes | 1.32***(1.26, 1.39) | 1.10***(1.04, 1.17) |
| *Country* | | |
| Angola 2015–2016 | 29.01***(21.69, 38.80) | 15.06***(11.15, 20.34) |
| Burkina Faso 2010 | 9.8***(7.36, 13.18) | 2.00***(1.48, 2.71) |
| Burundi 2016–2017 | 2.47***(1.84, 3.32) | 0.65**(0.48, 0.88) |
| Cameroon 2018 | 10.36***(7.73, 13.89) | 4.77***(3.52, 6.47) |
| Chad 2014–2015 | 64.14***(47.70, 86.24) | 15.62***(11.47, 21.28) |
| Comoros 2012 | 5.14***(3.75, 7.04) | 1.68***(1.21, 2.34) |
| Congo 2011–2012 | 3.96***(2.94, 5.34) | 1.37***(1.01, 1.87) |
| Congo DR 2013–2014 | 7.41***(5.53, 9.91) | 2.02***(1.50, 2.73) |
| Cote d'Ivoire 2011–2012 | 19.96***(14.89, 26.75) | 5.51***(4.06, 7.46) |
| Ethiopia 2016 | 44.08***(32.93, 59.01) | 12.64***(9.33, 17.11) |
| Gabon 2012 | 2.21***(2.00, 2.45) | 2.43***(1.78, 3.29) |
| Gambia 2013 | 5.86***(4.36, 7.38) | 2.42***(1.78, 3.29) |
| Ghana 2014 | 10.82***(4.05, 7.38) | 6.45***(4.75, 8.76) |
| Guinea 2018 | 22.90***(17.09, 30.69) | 5.60***(4.13, 7.59) |
| Kenya 2014 | 19.41***(14.50, 25.98) | 9.53***(7.05, 12.88) |
| Lesotho 2014–2015 | 6.57***(4.08, 9.13) | 6.03***(4.27, 8.52) |
| Liberia 2013 | 5.50***(4.08, 7.42) | 2.20***(1.61, 2.99) |
| Malawi 2015–2016 | 2.32***(1.73,3.12) | 0.67*(0.50, 0.91) |
| Mali 2018 | 12.10***(9.02, 16.21) | 3.18***(2.34, 4.31) |
| Mozambique 2011 | 16.45***(12.29, 22.02) | 6.90***(5.09, 9.33) |
| Namibia 2013 | 3.87***(2.85, 5.24) | 3.00***(2.19, 4.10) |
| Nigeria 2018 | 24.74***(18.52, 33.05) | 14.48***(10.73, 19.54) |
| Rwanda 2014–2015 | 2.36***1.74, 3.20) | 0.87(0.63, 1.19) |
| Sierra Leone 2019 | 4.15***(3.08, 5.58) | 1.49***(1.10, 2.03) |
| South Africa 2016 | Ref | Ref |
| Tanzania 2015–2016 | 14.11***(10.53, 18.90) | 5.56***(4.11, 7.53) |
| Togo 2013–2014 | 23.35***(17.42, 31.30) | 8.69***(6.41, 11.79) |
| Uganda 2016 | 8.61***(6.44, 11.53) | 3,41***(2.52, 4.61) |
| Zambia 2013–2014 | 6.49***(4.84, 8.71) | 2.69***(1.98, 3.64) |

(*Continued*)

**Table 3.** (Continued)

| Variables | Model 1 | Model 2 |
|---|---|---|
| | Odds Ratio | Adjusted Odds Ratio |
| | (95% Confidence interval) | (95% Confidence interval) |
| Zimbabwe 2015 | 5.22***(3.87, 7.04) | 3.84***(2.82, 5.22) |

*P<0.05

**p<0.01

***p<0.001

Ref = Reference category; Mean vif = 2.01

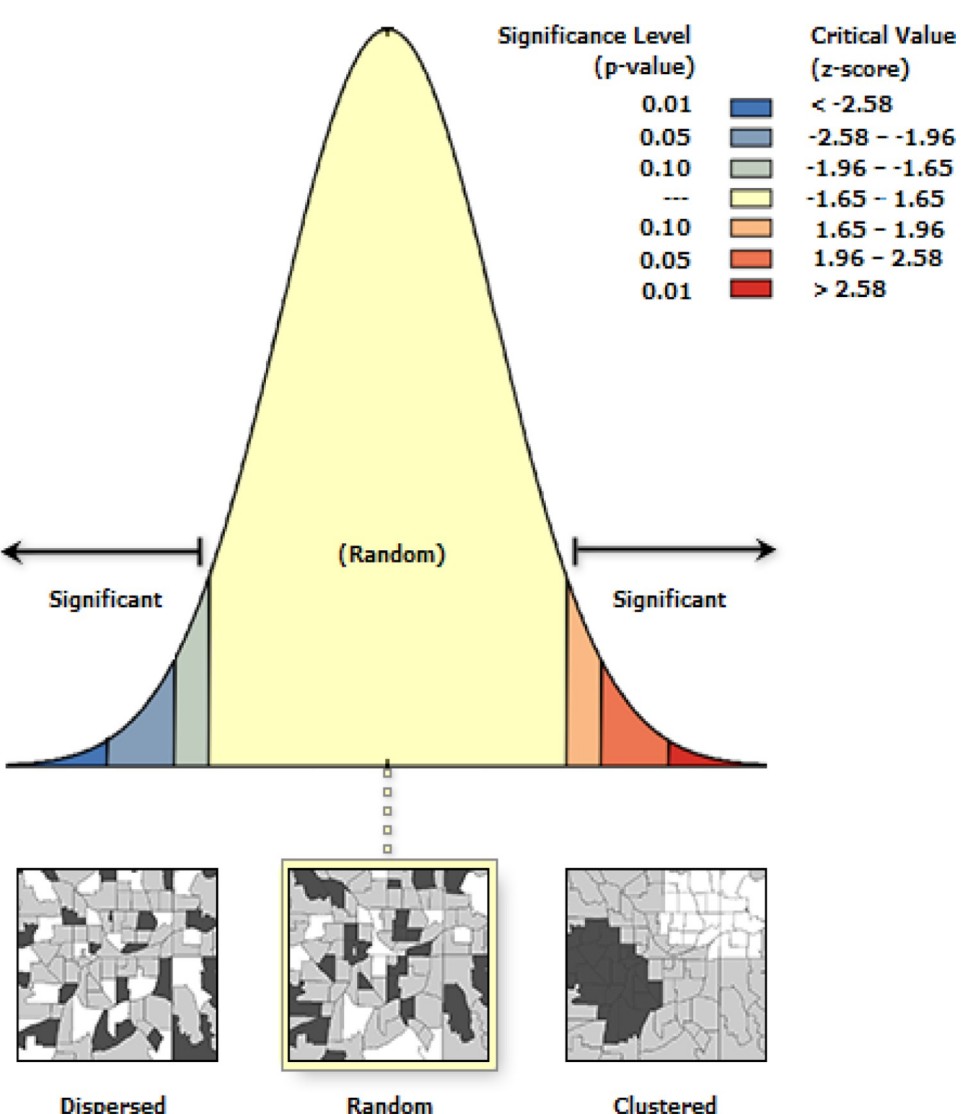

**Fig 1. Spatial autocorrelation analysis of unskilled birth attendance in sub-Saharan Africa.** Source: Authors' construct.

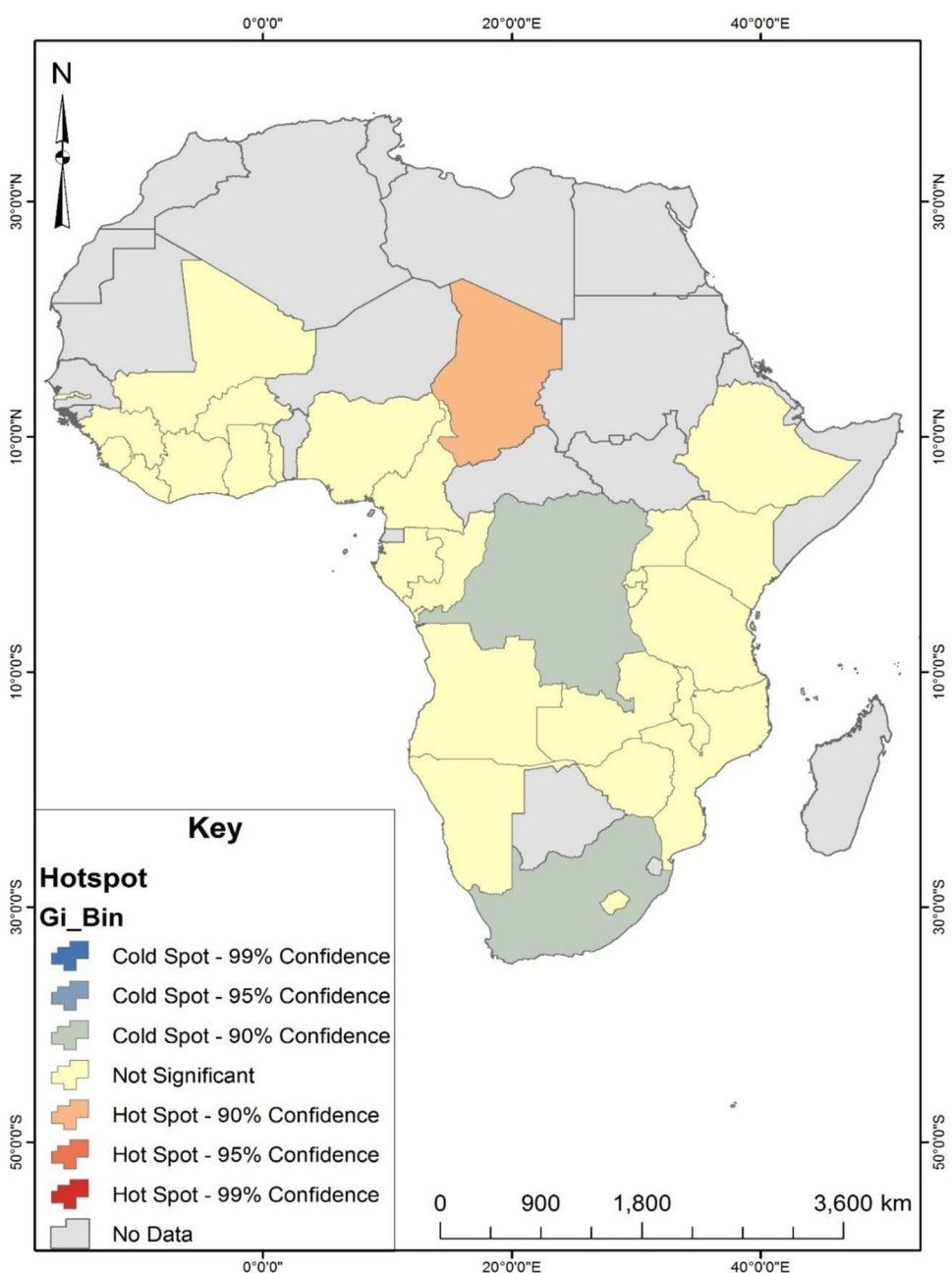

**Fig 2. Hotspot analysis of unskilled birth attendance in sub-Saharan Africa.** Source: Authors' construct.

hypothesis which states that the spatial distribution of unskilled birth attendance is not randomised was not accepted. Thus, the occurrence of unskilled birth attendance in SSA is by chance.

Although the occurrence of unskilled birth attendance was found to be random, the Hotspot Analysis (Fig 2) showed that at a 90% confidence level, the occurrence of unskilled birth attendance was likely to be found in Chad compared to the other selected countries. This implies a higher tendency for women in Chad to deliver without a skilled birth attendant than in the other selected countries. Also, two countries were found to have coldspots at a 90%

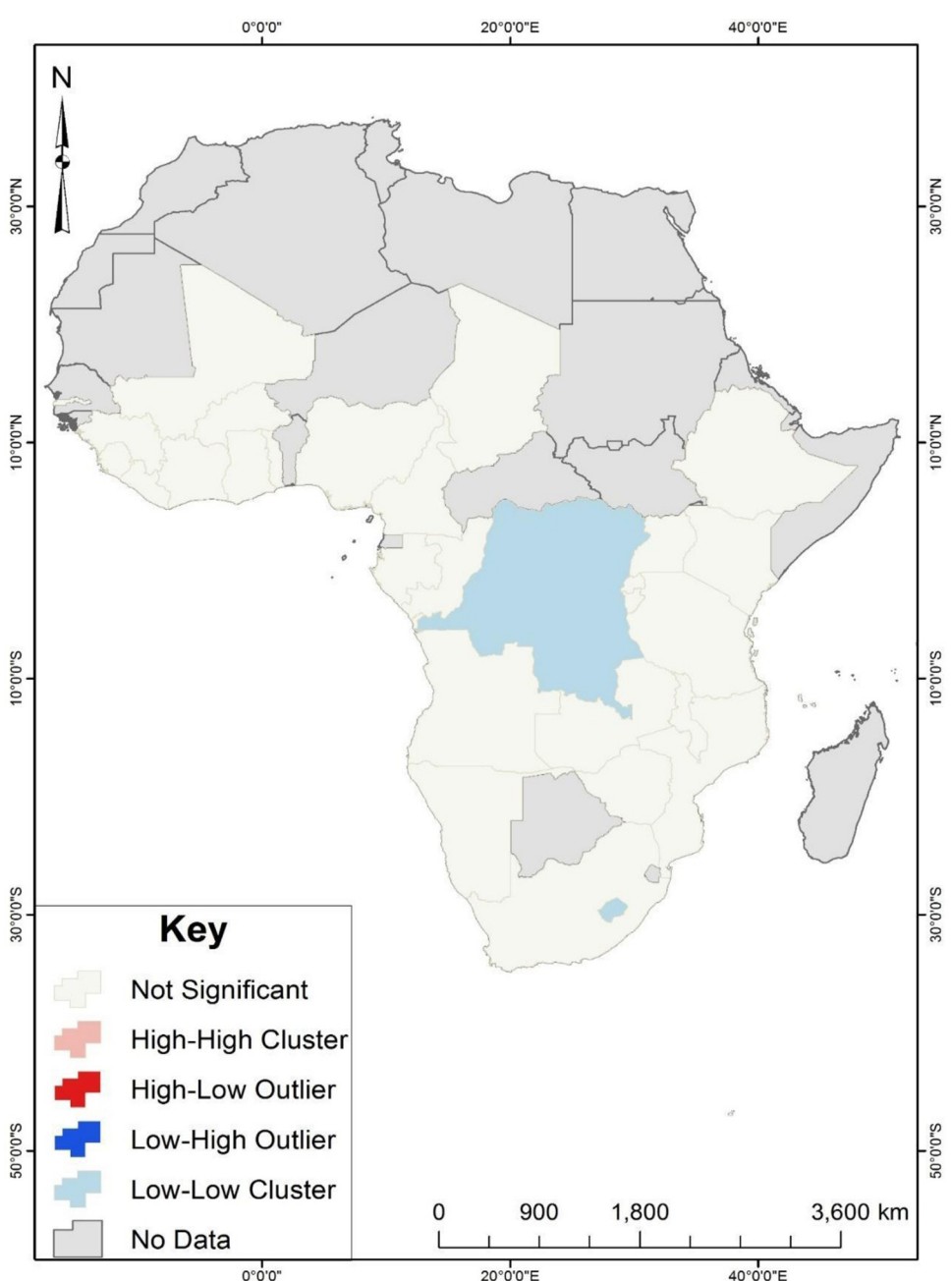

**Fig 3. Cluster and outlier analysis of unskilled birth attendance in sub-Saharan Africa.** Source: Authors' construct.

confidence level (Fig 2). These countries were South Africa and the Democratic Republic of Congo. Concerning all the selected countries for this study, the occurrence of unskilled birth attendance in South Africa and the Democratic Republic of Congo was found to be less likely.

Results from the cluster and outlier analysis (Fig 3) showed no high-high, high-low and low-high clusters. The cluster and outlier analysis showed that the Democratic Republic of Congo and Lesotho had low unskilled birth attendance and were surrounded by countries with low unskilled birth attendance. This is evident from the hotspot analysis presented in Fig 3.

## Discussion

This study examined levels of unskilled birth attendance among women in sub-Saharan Africa (SSA) and the factors associated with the utilisation of such services. The study further examined hot and cold spot countries in SSA where women utilised pregnancy and child delivery services from unskilled birth attendants. Consistent with a previous study [17], we found that women in rural areas had a higher likelihood of utilising services of unskilled birth attendants compared to women in urban areas. This is probably because rural areas often have a relatively large presence of unskilled birth attendants and have a higher likelihood of supporting customary child delivery practices compared to urban areas, such as keeping or disposing of a mother's placenta at a preferred location after delivery [20, 21].

Findings that women with the poorest wealth status as well as those who had a big problem getting money needed for care were more likely to utilise the services of unskilled birth attendants compared to those with the richest wealth status, as well as those who had no big problem getting money needed for care, demonstrate the important role played by financial capital in promoting the use of skilled birth attendants in SSA. In line with findings in a survey involving 80 low and middle-income countries across the world [22], our findings imply that some women utilised the services of unskilled birth attendants due to a lack of adequate capital to afford pregnancy and child delivery services from a skilled birth attendant. These findings show the need to establish or strengthen stimulus packages, such as free or low service costs in government facilities, particularly, for pregnant women in low-income settings [23]. However, measures should be taken to prevent such stimulus packages from becoming a motivation for proliferation in childbirth.

The findings further showed that teenage pregnancy (pregnancy among women below 20 years old) is a risk factor for the utilisation of services from unskilled birth attendants and this confirms a similar study indicating that early age at first childbirth is associated with low-skilled assistance during delivery [24]. Multiple reasons may account for this result including shame or stigma associated with teenage pregnancy in many African settings, financial constraints as many teenagers are in school and thus have no or low income to afford the services of a skilled birth attendant, and absence of partner support during pregnancy as teenagers are likely to be unmarried in their pregnancy [25–27].

Moreover, we found that women with low levels of education as well as those who were less exposed to media messages were more likely to utilise the services of unskilled birth attendants compared to highly educated women and those exposed to mass media. A possible reason for these findings is that educated women are likely to be more knowledgeable about the risks associated with utilising services of unskilled birth attendants and the mass media, particularly, the print media, is an important source of maternal health information for educated women compared with women who are less educated [28]. Additionally, educated women are more likely to be financially empowered than less educated women [29], which can enable them to afford skilled maternal healthcare services compared to less educated women. Our findings correspond with a study conducted in Nigeria [30] and indicate that promoting formal education among women in SSA is an important indirect approach to reducing the risk of child delivery supervised by an unskilled birth attendant in the region.

Concerning the issue of distance and clinical attendance, the findings showed that women who saw distance to a healthcare facility as a big problem as well as those who had less than 4 antenatal care visits had a higher likelihood of utilising the services of unskilled birth attendants during delivery compared to women who stated that distance to healthcare facilities was not a big problem as well as those who had more than 4 antenatal care visits. These findings are consistent with previous country-specific studies in SSA [17, 31–33]. It is important to be

reminded that embedded in the problem of long distance to healthcare facilities are other possible challenges such as time and financial costs associated with utilising the services of skilled birth attendants in formal healthcare settings as well as lack of available transport to a healthcare facility, especially, in rural settings and during late hours of the night [31]. Together, these distance-related factors may have influenced the women to utilise the services of unskilled birth attendants in the region.

Our findings indicate that women who had ever experienced the demise of their under-five children and those who had desired to have children later were likely to utilise services from unskilled birth attendants. These results are quite difficult to explain due to a lack of adequate published papers on these specific variables. Nevertheless, the findings may imply that many women who had lost their under-five children in previous deliveries as well as those who were not desiring an immediate child had no alternatives to services from unskilled birth attendants due to some preceding factors highlighted in this study, such as long distance to healthcare facilities, low-income status, poor maternal education, among others.

Overall, the findings indicate the need to amplify education for women in SSA on the benefits of seeking professional care and suggest a call for improvements in the psychosocial interactions between healthcare professionals and women in SSA. They also show a reflection of outcomes from previous studies focused on skilled birth attendance among women in SSA [5, 13–16]. The gross disparities in unskilled birth attendance among the various characteristics of women in this present study suggest that socio-economic inequalities among women are strong determinants of their health. Many SSA countries have fragile healthcare systems with a lack of skilled birth attendants and aggravated by an increased cost of medical care, geographic inaccessibility of healthcare facilities, among others [34]. Hence, many socially disadvantaged women, such as those with no or low exposure to media advocacies on maternal health, low levels of education, low incomes, and living in rural areas, are likely to continue using the relatively cheaper and easily accessible services of unskilled birth attendants.

The outcomes from our geospatial analysis further suggest significant spatial variations in unskilled birth attendance among women in the thirty (30) SSA countries, but those variations did not occur by chance. This means that although the use of services from unskilled birth attendants during pregnancy and delivery in SSA is a holistic problem, the situation varies by country of residence. Notably, with an outstanding level of unskilled birth attendance (69%), Chad can be classified as a hotspot and a high-risk country for unskilled birth attendance compared to its neighbouring countries. A possible reason is that the country has a relatively high incidence of unskilled birth attendance compared to its neighbouring countries. On the other hand, South Africa (3%) can also be classified as a coldspot country where women predominantly seek maternal health care from skilled birth attendants. The findings indicate that even though the geospatial distributions of unskilled birth attendance were random, there appeared to be some notable geospatial variations in unskilled birth attendance among women in some parts of SSA ranging from hotspots to coldspots as observed by [17]. The government of Chad and concerned healthcare institutions in the country need to seriously consider developing urgent interventions to address the high rate of unskilled birth attendance among women in the country.

## Strengths and limitations

This study is characterised by a few strengths. The geospatial aspect of the analysis provided a novel and deeper insight into the context of unskilled birth attendance in sub-Saharan Africa (SSA). To the best of our knowledge, this is also the first study on unskilled birth attendance that has employed nationally representative samples drawn from thirty (30) SSA countries. We

used a large dataset comprising 200,736 women aged between 15–49 years which increased the statistical power of our analysis. Nevertheless, the study is constrained by some shortcomings. First, it is worth noting that many of the surveys were collected in different periods which may impact the findings due to possible changes in outcomes over time as many sub-Saharan African countries do not collect their DHS data in the same year and at the same period. Second, some sub-Saharan African countries were excluded from the study because they lacked data on the outcome variable. Third, the outcome was self-reported and there is a possibility of social desirability and recall bias in the responses as women from cultural orientations where utilisation of unskilled birth services is stigmatised may provide favourable responses to conform to the ethos of their cultural orientations. Lastly, contextual factors such as cultural norms and attitudes of health service providers would have been interesting to examine in this study, but such variables were not captured in the DHS dataset.

## Implications for policy and practice

The multiple factors associated with unskilled birth attendance among the women in this study imply that there is a need for multi-faceted and local-specific policies and programmes in addressing the problem of unskilled birth attendance in SSA. Thus, a "one size fits all" approach may not be efficient in addressing unskilled birth attendance among women in the region. Some possibly useful strategies may include the provision of routine localised or rural health education for pregnant women about the importance of utilising the services of skilled birth attendants. Governments in SSA need to also make it a priority to extend healthcare facilities to rural areas to bridge the distance gap and improve the accessibility of skilled birth services for pregnant women in disadvantaged settings.

It is also important to recognise that although there is a general advocate for skilled birth attendance, there are some situations where service use from unskilled birth attendants may be non-negotiable. For instance, access to healthcare can be challenging in many conflict-affected countries or areas in protracted armed conflicts where healthcare workers have vacated their posts due to fear of persecution [3]. Some regions in SSA also have significant scarcities in skilled birth attendants making access to professional care limited [9]. As revealed in this study, long-distance to healthcare facilities, rural residency, and low-income levels can also deter women from accessing professional care during pregnancy and child delivery. In such circumstances, health policy advocates, local governments, and concerned healthcare institutions can collaborate to develop a system whereby unskilled birth attendants are well-trained and equipped with the needed modern equipment to facilitate optimal care for pregnant women. In recent times where telehealth and telemedicinal approaches are emerging as useful resources, the role played by unskilled birth attendants can be integrated into local healthcare systems and adequately monitored. A practical example is a situation in Timor-Leste where unskilled birth attendants were incorporated into a family health promoter programme which played a crucial role in delivering and increasing access to reproductive health services in rural communities of the nation. Although it required long-term dedication and effective collaborations, the current reduction in maternal mortality ratio in Timor-Leste is encouraging and serves to illustrate how such approaches can be useful [35].

## Conclusions

This study has confirmed that many women in sub-Saharan Africa (SSA) continue to utilise the services of unskilled birth attendants during pregnancy and delivery. The study also concludes that women in rural areas, with low income, aged below 20 years, with no or low levels of education, less exposed to mass media information, distanced away from healthcare

facilities, and with low levels of antennal attendance (<4 hospitals or clinic attendance) are likely to utilise the services of unskilled birth attendants. These findings suggest that multiple interventions focused on these contextual factors are needed in SSA. We propose an integrated approach where skilled birth attendants are prioritised, but in situations where there are shortages of skilled birth attendants, unskilled birth attendants can be trained, licensed, and monitored to save the lives of pregnant women. Extending maternal education and healthcare facilities or centres to rural areas should be a priority for managers of healthcare systems in SSA. Intensifying education for pregnant women about the importance of using skilled or professional healthcare during pregnancy, delivery, and post-delivery periods is also needed. Future studies can enhance a deeper understanding of these results by exploring our findings from qualitative perspectives.

## Acknowledgments

We acknowledge Measure DHS for providing us with the data upon which the findings of this study were based.

## Author Contributions

**Conceptualization:** Isaac Yeboah Addo, Evelyn Acquah, Samuel H. Nyarko, Kwamena Sekyi Dickson.

**Data curation:** Kwamena Sekyi Dickson.

**Formal analysis:** Ebenezer N. K. Boateng, Kwamena Sekyi Dickson.

**Methodology:** Samuel H. Nyarko.

**Supervision:** Kwamena Sekyi Dickson.

**Validation:** Isaac Yeboah Addo, Samuel H. Nyarko.

**Visualization:** Ebenezer N. K. Boateng.

**Writing – original draft:** Isaac Yeboah Addo, Evelyn Acquah, Samuel H. Nyarko, Ebenezer N. K. Boateng, Kwamena Sekyi Dickson.

**Writing – review & editing:** Isaac Yeboah Addo, Evelyn Acquah, Samuel H. Nyarko, Ebenezer N. K. Boateng, Kwamena Sekyi Dickson.

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
