## [Decision Letter · Decision Letter 0]

5 Jul 2022

PONE-D-21-36273Factors associated with unskilled birth attendance among women in Sub-Saharan Africa: a multilevel geospatial analysis of demographic and health surveysPLOS ONE

Dear Dr. Dickson,

Thank you for submitting your manuscript to PLOS ONE. After careful consideration, we feel that it has merit but does not fully meet PLOS ONE’s publication criteria as it currently stands. Therefore, we invite you to submit a revised version of the manuscript that addresses the points raised during the review process.

We look forward to receiving your revised manuscript.

Kind regards,

Tsegaye Lolaso Lenjebo, MPH

Academic Editor

PLOS ONE

Journal Requirements:

2. PLOS requires an ORCID iD for the corresponding author in Editorial Manager on papers submitted after December 6th, 2016. Please ensure that you have an ORCID iD and that it is validated in Editorial Manager. To do this, go to ‘Update my Information’ (in the upper left-hand corner of the main menu), and click on the Fetch/Validate link next to the ORCID field. This will take you to the ORCID site and allow you to create a new iD or authenticate a pre-existing iD in Editorial Manager. Please see the following video for instructions on linking an ORCID iD to your Editorial Manager account: https://www.youtube.com/watch?v=_xcclfuvtxQ.

4. Please include your tables as part of your main manuscript and remove the individual files. Please note that supplementary tables (should remain/ be uploaded) as separate "supporting information" files.

5. We note that Figures 2 &3 in your submission contain [map/satellite] images which may be copyrighted. All PLOS content is published under the Creative Commons Attribution License (CC BY 4.0), which means that the manuscript, images, and Supporting Information files will be freely available online, and any third party is permitted to access, download, copy, distribute, and use these materials in any way, even commercially, with proper attribution. For these reasons, we cannot publish previously copyrighted maps or satellite images created using proprietary data, such as Google software (Google Maps, Street View, and Earth). For more information, see our copyright guidelines: http://journals.plos.org/plosone/s/licenses-and-copyright.

 a. You may seek permission from the original copyright holder of Figures 2 & 3 to publish the content specifically under the CC BY 4.0 license. 

Reviewers' comments:

Reviewer's Responses to Questions

**Comments to the Author**

1. Is the manuscript technically sound, and do the data support the conclusions?

Reviewer #1: Yes

Reviewer #2: Yes

2. Has the statistical analysis been performed appropriately and rigorously? 

Reviewer #1: Yes

Reviewer #2: Yes

3. Have the authors made all data underlying the findings in their manuscript fully available?

Reviewer #1: Yes

Reviewer #2: Yes

4. Is the manuscript presented in an intelligible fashion and written in standard English?

Reviewer #1: No

Reviewer #2: Yes

5. Review Comments to the Author

Reviewer #1: The manuscript entitled “Factors associated with unskilled birth attendance among women in Sub-Saharan Africa: a multilevel geospatial analysis of demographic and health surveys” was reviewed carefully. There are some major and minor issues that should be considered.

Abstract: please consider to limit the abstract to 250 words. Background section is longer than other parts. The result section of abstract is not supported by Effect size. Please revise this section.

Method :

Please indicate confounders that the model was adjusted for.

Line 153-154: both lines are not essential. H1 is enough if authors insist in mentioning it in this section.

Result:

Line 171: It is not common to start the results with tables at first. It is suggested to revise this line.

Line 192-193: same as above.

Line 199-206: It is suggested to re-analysis the variables (making variables reverse) to have risk factor instead of preventive factor for better interpretation and understanding of readers.

Please indicate the confounders which were adjusted in this model.

Line236: unlike other analysis , authors selected 90% confidence interval for this analysis. Please give some reasons for this discrepancy.

Tables:

Please indicate percentage with 95% CI.

Discussion:

Line 2: 254-262 : the structure of scientific writing is not suitable for this part and is much more related to introduction. Please consider that the first paragraph of discussion should answer the research question of the introduction based on the eye-catching results of this study. Please revise.

The main key home massage of this study which makes this distinguish from other local mentioned published articles that authors had mentioned in introduction is not crystal clear in this manuscript. Please discuss it with more prominent evidence.

Reviewer #2: I found the paper good. The finding is supported by the analysis technique. It covers large geographical areas and it generates valuable information for public health authorities in the study areas. There are some grammatical errors and editorial issues that are highlighted yellow in the document. However, there is similar topic done in Ghana by similar authors and I have raised a question regarding this issue in my comments and suggestions that I will attach to you.

6. PLOS authors have the option to publish the peer review history of their article (what does this mean?). If published, this will include your full peer review and any attached files.

Reviewer #1: No

Reviewer #2: No

---

## [Author Response · Author response to Decision Letter 0]

12 Sep 2022

PONE-D-21-36273

Factors associated with unskilled birth attendance among women in Sub-Saharan Africa: a multilevel geospatial analysis of demographic and health surveys

PLOS ONE

Dear Prof. Tsegaye Lolaso Lenjebo,

We are grateful to you and the reviewers for your comments on our paper entitled " Factors associated with unskilled birth attendance among women in Sub-Saharan Africa: a multilevel geospatial analysis of demographic and health surveys". We would also take this opportunity to thank the reviewers for finding merit in this paper and suggesting that it can be suitable for publication in your renowned journal if we incorporate some major revisions. We have taken notice of all the comments raised by the reviewers and have responded accordingly as follows. Please be informed that the reviewers' comments are in black texts whereas our responses are in red texts. 

Responses to the Editor’s Comments

Response: Thank you. We have included a separate caption for each figure as recommended.

4. Please include your tables as part of your main manuscript and remove the individual files. Please note that supplementary tables (should remain/ be uploaded) as separate "supporting information" files.

Response: Thank you. We have included the tables as part of the main manuscript and removed the individual files as recommended

5. We note that Figures 2 &3 in your submission contain [map/satellite] images which may be copyrighted. All PLOS content is published under the Creative Commons Attribution License (CC BY 4.0), which means that the manuscript, images, and Supporting Information files will be freely available online, and any third party is permitted to access, download, copy, distribute, and use these materials in any way, even commercially, with proper attribution. For these reasons, we cannot publish previously copyrighted maps or satellite images created using proprietary data, such as Google software (Google Maps, Street View, and Earth). For more information, see our copyright guidelines: http://journals.plos.org/plosone/s/licenses-and-copyright.

 a. You may seek permission from the original copyright holder of Figures 2 & 3 to publish the content specifically under the CC BY 4.0 license. 

Response: We can confirm that Figures 1-3 were created or constructed by the authors based on the study data using ArcMap version 10.5. Permission from a copyright holder is not applicable in this context. We have, therefore, provided a source in the manuscript showing that the Figures were self-created by the authors.

Responses to the Reviewers' comments:

Reviewer #1: The manuscript entitled “Factors associated with unskilled birth attendance among women in Sub-Saharan Africa: a multilevel geospatial analysis of demographic and health surveys” was reviewed carefully. There are some major and minor issues that should be considered.

Response: Thank you. We have addressed the concerns as follows.

Abstract: please consider to limit the abstract to 250 words. Background section is longer than other parts.

Response: Thank you. We have reduced the abstract to 227 words and have cut down the background section as recommended.

The result section of abstract is not supported by Effect size. Please revise this section.

Response: Thank you. This section has been revised.

Method :

Please indicate confounders that the model was adjusted for.

Response: Thank you. The section under the analytical procedure has been revised as recommended.

Line 153-154: both lines are not essential. H1 is enough if authors insist in mentioning it in this section.

Response

Thank you for the comment. H0 has been deleted accordingly.

Result:

Line 171: It is not common to start the results with tables at first. It is suggested to revise this line.

Response

The entire sentence has been deleted. Thank you for your observation 

Line 192-193: same as above.

Response

The entire sentence has been deleted. Thank you for your observation 

Line 192-193: same as above.

Response The entire sentence has been deleted. Thank you for your observation

Line 199-206: It is suggested to re-analysis the variables (making variables reverse) to have risk factor instead of preventive factor for better interpretation and understanding of readers.

Response: Thank you. This section has been re-analysed and revised as suggested. 

Please indicate the confounders which were adjusted in this model.

Response Thank you. This has been added. Please see the “Analytical procedure” section under the Methods.

Line 236: unlike other analysis, authors selected 90% confidence interval for this analysis. Please give some reasons for this discrepancy.

Response: Thank you for this observation. The tool used for running the Getis Ord G spatial analysis generates a seven-level categorised output based on confidence level. These seven-level categorised outputs are segmented into three, coldspot (99%, 95% and 90%), random and hotspot (99%, 95% and 90%). Our output automatically revealed that the distribution of unskilled birth attendance is spatially and significantly clustered at 90% confidence level. 

Unlike the other models where we had the liberty to select 95% confidence interval levels, we did not have control over the confidence interval for the geospatial analysis as the software automatically selected the 90% confidence interval based on the level of significance within the spatial distribution clusters. 

Tables:

Please indicate percentage with 95% CI.

Response: Please be informed that the Footnotes in Table 3 show the meaning of the * in the table. Thus, *p<0.05 equals 95% CI

Discussion:

Line 2: 254-262 : the structure of scientific writing is not suitable for this part and is much more related to introduction. Please consider that the first paragraph of discussion should answer the research question of the introduction based on the eye-catching results of this study. Please revise.

Response Thank you. The structure has been revised as recommended

The main key home massage of this study which makes this distinguish from other local mentioned published articles that authors had mentioned in introduction is not crystal clear in this manuscript. Please discuss it with more prominent evidence.

Response Thank you. The Discussion section of the manuscript has been revised to show the key variables and the take-home message. 

Reviewer #2: I found the paper good. The finding is supported by the analysis technique. It covers large geographical areas and it generates valuable information for public health authorities in the study areas. 

Response: Thank you for this positive feedback.

There are some grammatical errors and editorial issues that are highlighted yellow in the document.

Response: Thank you for the highlights. The issues have been resolved.

However, there is similar topic done in Ghana by similar authors and I have raised a question regarding this issue in my comments and suggestions that I will attach to you.

Response: Thank you. We have explained this in the main comment. 

Comments and suggestions

Line-38: “older current age and age at birth”

• The association direction for age at birth is not specified.

• Does the word “older” applies for both or not? If it applies for both then better if you write separately as older current age, older age at birth …

Response Thank you for the observation. The results section of the abstract has been revised.

Line 38 and 39:

• The two variables; rich wealth status Vs not having money problems for healthcare

• Both are complementary to each other. So, I recommend to select and use one. Wealth status is better.

• Response Thank you for the observation. The results section of the abstract has been revised.

Line 43:

• Why Burundi?

• Lowest proportion is from South Africa (SA) (less exposed to the outcome; meaning less proportion of the outcome is indicated from the descriptive analysis. So, I recommend to make SA as reference)

• Then what is the implication behind the result? All counties have higher odds? 

Response: Thank you. The analysis has been redone and South Africa has been made the reference category. The section has been revised.

Line 78-79:

• Add something about the spatial pattern of the issue in addition to what you stated.

Response: Thank you. An additional sentence has been included to show that there are limited studies utilising spatial analysis to explore the distribution of unskilled birth attendance in SSA.

Line 93:

• The word “Explored” is not the appropriate action verb for such type of quantitative analysis. Better to use “identified” or “examined” …

Response: Thank you for this observation. The word “explored” has been replaced with “examined”

Line 95: 

• The same comment as line 93 (Exploring???)

Response. Thank you. The word “Exploring” has been replaced with “Examining”

• Line 101: 

• Could it be different for countries? If it the year is different how you handle the analysis? How do you control the effect of time? So, it is good if you make DHS data sets which are produced in the same year across different countries. Otherwise, the result is biased due to time effect or make it time series analysis?

• Response: Thank you for this important observation. We agree with the reviewer that the differences in survey years may affect the results due to possible effect of time. However, DHS data for sub-Saharan African countries are commonly collected in different years and at different periods. This has been a notable limitation of most studies using pooled DHS data of various countries. As a result, we have added this as one of the limitations of the study. 

Line 180-187:

• The description is not as such important as far as these variables are included in the regression model. The proportion (percentage) here is crude and it doesn’t add any value. So, delete it and try to include all variables into the model and report their adjusted effect on the outcome. 

Response: Thank you for this comment. While we appreciate the reviewer’s suggestion, in this instance, we plead to disagree. The description of participants is commonly provided in this type of public health research to give readers information on the number of respondents in each variable category. It also sets the basis for interpreting the results (particularly, in the model) and helps in identifying possible limitations induced by small samples in the variable categories. 

Line 194:

• Make is showed i.e.) add “ed”

Response: This has been rectified. Thank you

Line 202: 

• Delete (Table 3).; as it is reported at the end of the paragraph.

Response The expression “Table 3” has been deleted in the line as suggested. Thank you.

Line 213: 

• Delete the word “see” as it is not important (be consistent).

Response. Thank you. The word “see” has been deleted as recommended.

Line 222: 

• A dd (Table 3) immediately after the word child.

Response: Thank you. The expression “Table 3” has been added as suggested.

Line 254-258:

• The paragraph you stated as a starting point and discussion is not directly addressed through this analysis. So, I recommend to write based on the study objectives. This study didn’t address the effect of unskilled birth attendance on maternal death though it is addressed by other studies. So, try to focus on the study objectives. This study has nothing related to this issue.

• Start with your main finding and then try to compare to others studies and policy implications

• The proportion of the outcome is not discussed well. so, discuss the percentage in relation to expected proportion based on global initiatives

Response: Thank you. We acknowledge this, have deleted the part not related to the analysis and have revised the Discussion section. 

Line 263-265

• Delete it; what is the important? Or make it short (paraphrase it). 

Response: Thank you. This has been done.

• Line 266-280: 

• It is good you have stated some important statement for discussion. But these significant variables must be discussed separately with their corresponding justification. E.g.) why younger women higher odds of utilizing the service? Labor and child birth experience Vs Woman’s age and then Vs unskilled birth attendance?

• So, the same is true for other variables found to be significant in the final model need to be discussed. 

Response: Thank you. The discussion has been revised as recommended.

Line 283:

• The use of conjunction (“Interestingly,”) is not correct for under five demise Vs Unskilled birth attendance. 

• The reasoning stated in the document is not convincing. So, try to make it informative and convincing? Imogie (it is single study) and it is difficult to compare this study where it is the analysis many countries with a single study in Nigeria?

Response

● The word interestingly has been deleted. The discussion has been revised as recommended. Thank you for the observation.

• Line 281-296:

• The reasoning or justification need to be modified. 

Response: Thank you. The discussion has been revised as recommended.

Line 297-307:

• What could be the reason for hotspot for some countries and coldspot for some countries? Justification if you can?

Response: The probable cause of hotspot could be due to the high incidence of unskilled birth attendance among neighbouring countries and vice versa. A further explanation has now been included in the manuscript.

• Line 309-316: regarding the strength

• Statements about strength of the study need to be paraphrased. Please come to the point in short (the only strength could be using large sample size; nothing else). Using logistic model can not be a strength. The spatial can be a strength. 

Response: The strength subsection has been revised as recommended. Thank you

Line 316-327:

• Please try to state your own study or analysis limitation. 

• And then what are the potential limitation that you will share with DHS protocol.

• Being cross-sectional can’t be a limitation for your analysis. (Causal analysis is not your plan/objective)

• Still this subsection needs to paraphrased. 

Response: This subsection has been revised as recommended by the reviewer.

Line 357: delete the “.” Before the reference

Response: Thank you. This has been done.

The points stated under “Implications for policy and practice” can be written under discussion and conclusion based on the variables you discussed. Under discussion you can finalize with its public health policy and implementation implications

Response Thank you for this suggestion. While we appreciate the reviewer’s suggestion, on this occasion, we plead to maintain the section on “Implications for policy and practice” as that section deals with high-level implications that are not specific to a particular variable.

🡺 Similar topic is done in Ghana by similar author; so, why you include Ghana in this analysis? What makes different from that study? What is new?

🡺 Response: While we acknowledge that a similar topic has been done for Ghana please be informed that only one out of the five current authors was part of the previous study conducted in Ghana. More importantly, we included Ghana in the analysis to enable us to directly compare the previous findings on Ghana (particularly the geospatial elements) to the remaining 29 Sub-Saharan African countries included in this study. In addition, the previous study on Ghana did not include the following important variables captured in our study: age at first birth, marital status, getting medical help for self: permission to go, getting medical help for self: getting money needed for treatment, desire for children, antenatal care visits, and under-five mortality.

The key legends for figure 2 and 3 Vs the colours in the figure are not comparable and this hide the information that can be captured from the figures.

Response: Thank you. Although the two figures may be comparing the distribution of unskilled birth attendance, the colours can be different since they are analysed by two different tools. Also, using the same colours for the two would not be possible because the colour grading used in figure 2 shows magnitude whereas that of figure 3 does not.

Thank you.

---

## [Editor Report · Decision Letter 1]

31 Oct 2022

PONE-D-21-36273R1Factors associated with unskilled birth attendance among women in Sub-Saharan Africa: a multilevel geospatial analysis of demographic and health surveysPLOS ONE

Dear Dr. Dickson,

Thank you for submitting your manuscript to PLOS ONE. After careful consideration, we feel that it has merit but does not fully meet PLOS ONE’s publication criteria as it currently stands. Therefore, we invite you to submit a revised version of the manuscript that addresses the points raised during the review process.

We look forward to receiving your revised manuscript.

Kind regards,

Kannan Navaneetham, PhD

Academic Editor

PLOS ONE
---

## [Author Response · Author response to Decision Letter 1]

3 Dec 2022

PONE-D-21-36273

Factors associated with unskilled birth attendance among women in Sub-Saharan Africa: a multilevel geospatial analysis of demographic and health surveys

PLOS ONE

Dear Prof. Tsegaye Lolaso Lenjebo,

We are grateful to you and the reviewers for your comments on our paper entitled " Factors associated with unskilled birth attendance among women in Sub-Saharan Africa: a multilevel geospatial analysis of demographic and health surveys". We would also take this opportunity to thank the reviewers for finding merit in this paper and suggesting that it can be suitable for publication in your renowned journal if we incorporate some major revisions. We have taken notice of all the comments raised by the reviewers and have responded accordingly as follows. Please be informed that the reviewers' comments are in black texts whereas our responses are in red texts. 

Responses to the Editor’s Comments

Response: Thank you. We have included a separate caption for each figure as recommended.

4. Please include your tables as part of your main manuscript and remove the individual files. Please note that supplementary tables (should remain/ be uploaded) as separate "supporting information" files.

Response: Thank you. We have included the tables as part of the main manuscript and removed the individual files as recommended

5. We note that Figures 2 &3 in your submission contain [map/satellite] images which may be copyrighted. All PLOS content is published under the Creative Commons Attribution License (CC BY 4.0), which means that the manuscript, images, and Supporting Information files will be freely available online, and any third party is permitted to access, download, copy, distribute, and use these materials in any way, even commercially, with proper attribution. For these reasons, we cannot publish previously copyrighted maps or satellite images created using proprietary data, such as Google software (Google Maps, Street View, and Earth). For more information, see our copyright guidelines: http://journals.plos.org/plosone/s/licenses-and-copyright.

 a. You may seek permission from the original copyright holder of Figures 2 & 3 to publish the content specifically under the CC BY 4.0 license. 

Response: We can confirm that Figures 1-3 were created or constructed by the authors based on the study data using ArcMap version 10.5. Permission from a copyright holder is not applicable in this context. We have, therefore, provided a source in the manuscript showing that the Figures were self-created by the authors.

Responses to the Reviewers' comments:

Reviewer #1: The manuscript entitled “Factors associated with unskilled birth attendance among women in Sub-Saharan Africa: a multilevel geospatial analysis of demographic and health surveys” was reviewed carefully. There are some major and minor issues that should be considered.

Response: Thank you. We have addressed the concerns as follows.

Abstract: please consider to limit the abstract to 250 words. Background section is longer than other parts.

Response: Thank you. We have reduced the abstract to 227 words and have cut down the background section as recommended.

The result section of abstract is not supported by Effect size. Please revise this section.

Response: Thank you. This section has been revised.

Method :

Please indicate confounders that the model was adjusted for.

Response: Thank you. The section under the analytical procedure has been revised as recommended.

Line 153-154: both lines are not essential. H1 is enough if authors insist in mentioning it in this section.

Response

Thank you for the comment. H0 has been deleted accordingly.

Result:

Line 171: It is not common to start the results with tables at first. It is suggested to revise this line.

Response

The entire sentence has been deleted. Thank you for your observation 

Line 192-193: same as above.

Response

The entire sentence has been deleted. Thank you for your observation 

Line 192-193: same as above.

Response The entire sentence has been deleted. Thank you for your observation

Line 199-206: It is suggested to re-analysis the variables (making variables reverse) to have risk factor instead of preventive factor for better interpretation and understanding of readers.

Response: Thank you. This section has been re-analysed and revised as suggested. 

Please indicate the confounders which were adjusted in this model.

Response Thank you. This has been added. Please see the “Analytical procedure” section under the Methods.

Line 236: unlike other analysis, authors selected 90% confidence interval for this analysis. Please give some reasons for this discrepancy.

Response: Thank you for this observation. The tool used for running the Getis Ord G spatial analysis generates a seven-level categorised output based on confidence level. These seven-level categorised outputs are segmented into three, coldspot (99%, 95% and 90%), random and hotspot (99%, 95% and 90%). Our output automatically revealed that the distribution of unskilled birth attendance is spatially and significantly clustered at 90% confidence level. 

Unlike the other models where we had the liberty to select 95% confidence interval levels, we did not have control over the confidence interval for the geospatial analysis as the software automatically selected the 90% confidence interval based on the level of significance within the spatial distribution clusters. 

Tables:

Please indicate percentage with 95% CI.

Response: Please be informed that the Footnotes in Table 3 show the meaning of the * in the table. Thus, *p<0.05 equals 95% CI

Discussion:

Line 2: 254-262 : the structure of scientific writing is not suitable for this part and is much more related to introduction. Please consider that the first paragraph of discussion should answer the research question of the introduction based on the eye-catching results of this study. Please revise.

Response Thank you. The structure has been revised as recommended

The main key home massage of this study which makes this distinguish from other local mentioned published articles that authors had mentioned in introduction is not crystal clear in this manuscript. Please discuss it with more prominent evidence.

Response Thank you. The Discussion section of the manuscript has been revised to show the key variables and the take-home message. 

Reviewer #2: I found the paper good. The finding is supported by the analysis technique. It covers large geographical areas and it generates valuable information for public health authorities in the study areas. 

Response: Thank you for this positive feedback.

There are some grammatical errors and editorial issues that are highlighted yellow in the document.

Response: Thank you for the highlights. The issues have been resolved.

However, there is similar topic done in Ghana by similar authors and I have raised a question regarding this issue in my comments and suggestions that I will attach to you.

Response: Thank you. We have explained this in the main comment. 

Comments and suggestions

Line-38: “older current age and age at birth”

• The association direction for age at birth is not specified.

• Does the word “older” applies for both or not? If it applies for both then better if you write separately as older current age, older age at birth …

Response Thank you for the observation. The results section of the abstract has been revised.

Line 38 and 39:

• The two variables; rich wealth status Vs not having money problems for healthcare

• Both are complementary to each other. So, I recommend to select and use one. Wealth status is better.

• Response Thank you for the observation. The results section of the abstract has been revised.

Line 43:

• Why Burundi?

• Lowest proportion is from South Africa (SA) (less exposed to the outcome; meaning less proportion of the outcome is indicated from the descriptive analysis. So, I recommend to make SA as reference)

• Then what is the implication behind the result? All counties have higher odds? 

Response: Thank you. The analysis has been redone and South Africa has been made the reference category. The section has been revised.

Line 78-79:

• Add something about the spatial pattern of the issue in addition to what you stated.

Response: Thank you. An additional sentence has been included to show that there are limited studies utilising spatial analysis to explore the distribution of unskilled birth attendance in SSA.

Line 93:

• The word “Explored” is not the appropriate action verb for such type of quantitative analysis. Better to use “identified” or “examined” …

Response: Thank you for this observation. The word “explored” has been replaced with “examined”

Line 95: 

• The same comment as line 93 (Exploring???)

Response. Thank you. The word “Exploring” has been replaced with “Examining”

• Line 101: 

• Could it be different for countries? If it the year is different how you handle the analysis? How do you control the effect of time? So, it is good if you make DHS data sets which are produced in the same year across different countries. Otherwise, the result is biased due to time effect or make it time series analysis?

• Response: Thank you for this important observation. We agree with the reviewer that the differences in survey years may affect the results due to possible effect of time. However, DHS data for sub-Saharan African countries are commonly collected in different years and at different periods. This has been a notable limitation of most studies using pooled DHS data of various countries. As a result, we have added this as one of the limitations of the study. 

Line 180-187:

• The description is not as such important as far as these variables are included in the regression model. The proportion (percentage) here is crude and it doesn’t add any value. So, delete it and try to include all variables into the model and report their adjusted effect on the outcome. 

Response: Thank you for this comment. While we appreciate the reviewer’s suggestion, in this instance, we plead to disagree. The description of participants is commonly provided in this type of public health research to give readers information on the number of respondents in each variable category. It also sets the basis for interpreting the results (particularly, in the model) and helps in identifying possible limitations induced by small samples in the variable categories. 

Line 194:

• Make is showed i.e.) add “ed”

Response: This has been rectified. Thank you

Line 202: 

• Delete (Table 3).; as it is reported at the end of the paragraph.

Response The expression “Table 3” has been deleted in the line as suggested. Thank you.

Line 213: 

• Delete the word “see” as it is not important (be consistent).

Response. Thank you. The word “see” has been deleted as recommended.

Line 222: 

• A dd (Table 3) immediately after the word child.

Response: Thank you. The expression “Table 3” has been added as suggested.

Line 254-258:

• The paragraph you stated as a starting point and discussion is not directly addressed through this analysis. So, I recommend to write based on the study objectives. This study didn’t address the effect of unskilled birth attendance on maternal death though it is addressed by other studies. So, try to focus on the study objectives. This study has nothing related to this issue.

• Start with your main finding and then try to compare to others studies and policy implications

• The proportion of the outcome is not discussed well. so, discuss the percentage in relation to expected proportion based on global initiatives

Response: Thank you. We acknowledge this, have deleted the part not related to the analysis and have revised the Discussion section. 

Line 263-265

• Delete it; what is the important? Or make it short (paraphrase it). 

Response: Thank you. This has been done.

• Line 266-280: 

• It is good you have stated some important statement for discussion. But these significant variables must be discussed separately with their corresponding justification. E.g.) why younger women higher odds of utilizing the service? Labor and child birth experience Vs Woman’s age and then Vs unskilled birth attendance?

• So, the same is true for other variables found to be significant in the final model need to be discussed. 

Response: Thank you. The discussion has been revised as recommended.

Line 283:

• The use of conjunction (“Interestingly,”) is not correct for under five demise Vs Unskilled birth attendance. 

• The reasoning stated in the document is not convincing. So, try to make it informative and convincing? Imogie (it is single study) and it is difficult to compare this study where it is the analysis many countries with a single study in Nigeria?

Response

● The word interestingly has been deleted. The discussion has been revised as recommended. Thank you for the observation.

• Line 281-296:

• The reasoning or justification need to be modified. 

Response: Thank you. The discussion has been revised as recommended.

Line 297-307:

• What could be the reason for hotspot for some countries and coldspot for some countries? Justification if you can?

Response: The probable cause of hotspot could be due to the high incidence of unskilled birth attendance among neighbouring countries and vice versa. A further explanation has now been included in the manuscript.

• Line 309-316: regarding the strength

• Statements about strength of the study need to be paraphrased. Please come to the point in short (the only strength could be using large sample size; nothing else). Using logistic model can not be a strength. The spatial can be a strength. 

Response: The strength subsection has been revised as recommended. Thank you

Line 316-327:

• Please try to state your own study or analysis limitation. 

• And then what are the potential limitation that you will share with DHS protocol.

• Being cross-sectional can’t be a limitation for your analysis. (Causal analysis is not your plan/objective)

• Still this subsection needs to paraphrased. 

Response: This subsection has been revised as recommended by the reviewer.

Line 357: delete the “.” Before the reference

Response: Thank you. This has been done.

The points stated under “Implications for policy and practice” can be written under discussion and conclusion based on the variables you discussed. Under discussion you can finalize with its public health policy and implementation implications

Response Thank you for this suggestion. While we appreciate the reviewer’s suggestion, on this occasion, we plead to maintain the section on “Implications for policy and practice” as that section deals with high-level implications that are not specific to a particular variable.

🡺 Similar topic is done in Ghana by similar author; so, why you include Ghana in this analysis? What makes different from that study? What is new?

🡺 Response: While we acknowledge that a similar topic has been done for Ghana please be informed that only one out of the five current authors was part of the previous study conducted in Ghana. More importantly, we included Ghana in the analysis to enable us to directly compare the previous findings on Ghana (particularly the geospatial elements) to the remaining 29 Sub-Saharan African countries included in this study. In addition, the previous study on Ghana did not include the following important variables captured in our study: age at first birth, marital status, getting medical help for self: permission to go, getting medical help for self: getting money needed for treatment, desire for children, antenatal care visits, and under-five mortality.

The key legends for figure 2 and 3 Vs the colours in the figure are not comparable and this hide the information that can be captured from the figures.

Response: Thank you. Although the two figures may be comparing the distribution of unskilled birth attendance, the colours can be different since they are analysed by two different tools. Also, using the same colours for the two would not be possible because the colour grading used in figure 2 shows magnitude whereas that of figure 3 does not.

Thank you.

---

## [Editor Report · Decision Letter 2]

13 Jan 2023

Factors associated with unskilled birth attendance among women in Sub-Saharan Africa: a multilevel geospatial analysis of demographic and health surveys

PONE-D-21-36273R2

Dear Dr. Dickson,

We’re pleased to inform you that your manuscript has been judged scientifically suitable for publication and will be formally accepted for publication once it meets all outstanding technical requirements.

Kind regards,

Kannan Navaneetham, PhD

Academic Editor

PLOS ONE
---

## [Editor Report · Acceptance letter]

24 Jan 2023

PONE-D-21-36273R2 

Factors associated with unskilled birth attendance among women in sub-Saharan Africa: a multivariate-geospatial analysis of demographic and health surveys 

Dear Dr. Dickson:

I'm pleased to inform you that your manuscript has been deemed suitable for publication in PLOS ONE. Congratulations! Your manuscript is now with our production department. 

Kind regards, 

on behalf of

Prof. Kannan Navaneetham 

Academic Editor

PLOS ONE